# HERV-Derived *Ervpb1* Is Conserved in Simiiformes, Exhibiting Expression in Hematopoietic Cell Lineages Including Macrophages

**DOI:** 10.3390/ijms22094504

**Published:** 2021-04-26

**Authors:** Ayumi Matsuzawa, Jiyoung Lee, So Nakagawa, Johbu Itoh, Mahoko Takahashi Ueda, Satomi Mitsuhashi, Yuta Kochi, Tomoko Kaneko-Ishino, Fumitoshi Ishino

**Affiliations:** 1Department of Epigenetics, Medical Research Institute, Tokyo Medical and Dental University (TMDU), Tokyo 113-8510, Japan; amatepgn@tmd.ac.jp (A.M.); jlee.epgn@mri.tmd.ac.jp (J.L.); 2Department of Molecular Life Science, Tokai University School of Medicine, 143 Shimokasuya, Isehara 259-1193, Kanagawa, Japan; so@tokai.ac.jp; 3Department of Pathology, School of Medicine, Tokai University, 143 Shimokasuya, Isehara 259-1193, Kanagawa, Japan; itohj@is.icc.u-tokai.ac.jp; 4Department of Genomic Function and Diversity, Medical Research Institute, Tokyo Medical and Dental University (TMDU), Tokyo 113-8510, Japan; mahoko.gfd@mri.tmd.ac.jp (M.T.U.); satomits.gfd@mri.tmd.ac.jp (S.M.); y-kochi.gfd@mri.tmd.ac.jp (Y.K.); 5Department of Nursing, School of Medicine, Tokai University, 143 Shimokasuya, Isehara 259-1193, Kanagawa, Japan

**Keywords:** an *Env*-derived *ERVPb1* gene, a simiiforme-specific gene, retrovirus-derived genes (*RVDs*), human genome, hemangioblast cells, hematopoietic cells, macrophages, human iPSCs, organoid generation

## Abstract

(1) Background: The *ERVPb1* gene in humans is derived from an envelope (*Env*) gene of a human endogenous retrovirus group, HERV-P(b). The *ERVPb1* gene reportedly has a conserved open reading frame (ORF) in Old World monkeys. Although its forced expression led to cell-fusion in an ex vivo cell culture system, like other *Env*-derived genes such as *syncytin-1* and -*2,* its mRNA expression is not placenta-specific, but almost ubiquitous, albeit being quite low in human tissues and organs, implying a distinct role for *ERVPb1*. (2) Methods: To elucidate the cell lineage(s) in which the ERVPb1 protein is translated in human development, we developed a novel, highly sensitive system for detecting HERV-derived proteins/peptides expressed in the tissue differentiation process of human induced pluripotent stem cells (iPSCs). (3) Results: We first determined that *ERVPb1* is also conserved in New World monkeys. Then, we showed that the ERVPb1 protein is translated from a uniquely spliced *ERVPb1* transcript in hematopoietic cell lineages, including a subset of macrophages, and further showed that its mRNA expression is upregulated by lipopolysaccharide (LPS) stimulation in primary human monocytes. (4) Conclusions: *ERVPb1* is unique to Simiiformes and actually translated in hematopoietic cell lineages, including a subset of macrophages.

## 1. Introduction

Approximately 8% of the human genome is comprised of long terminal repeat (LTR) retrotransposons/human endogenous retroviruses (HERVs) [1,2]. Although most of these have lost their original activity due to mutations, there still remain more than 10,000 HERVs that have open reading frames (ORFs) comprising >100 amino acids (aa) related to GAG-, POL-, and ENV-derived proteins/peptides [3,4]. Thus, these HERVs are potentially a large source of as yet unidentified retrovirus-derived (*RVD*) genes in the human genome, although it remains to be determined whether proteins/peptides are actually translated from such ORFs and thus play a role in human development.

For example, more than ten *RVD* genes have been identified so far in the human genome, such as paternally expressed 10 (*PEG10*) and eight other retrotransposon Gag-like genes (*RTL*) (also called sushi-ichi-related retrotransposon homologues (*SIRHs*)) [5,6,7]. These genes play essential roles in the placenta and/or brain. There are also three *Env*-derived genes, *syncytin-1* and *-2* and *suppressyn*, that play essential roles in the placenta: two *syncytins* that play a role in the formation of the syncytial layer by inducing the fusion of syncytiotrophoblast cells, and *suppressyn*, which has a role in the inhibition of the activity of *syncytin-1* but not *syncytin-2* [8,9,10]. The latter *Env*-derived genes were captured (or domesticated) from three different HERVs, HERV-W, -FRD, and -Fb1, approximately 25, >40, and 25 million years ago (MYA), respectively. Therefore, *syncytin-1* and *suppressyn* are Catarrhine (Old World monkey)-specific, while *syncytin-2* is a Simiiforme-specific gene [9,11]. In contrast, the former genes encode GAG- (and sometimes also POL-) like proteins exhibiting the highest homology to the LTR sushi-ichi retrotransposon (each at most 20~30% homology). However, it is reasonable to think that these genes were derived from extinct retrovirus(es), because they were acquired (or domesticated) in the therian or eutherian ancestors approximately 168 and 148 MYA, respectively [12]. These data suggest that a number of as yet unidentified *RVD* genes may have been accumulated in a step-by-step manner in the human genome as mammalian-specific as well as primate-specific genes.

Among the *Env*-derived genes, *ERVPb1* (also called *ERVP(b)* is reported to have an ORF comprising 665 amino acids in humans and retains several features typical of ENV: it is comprised of surface (SU) and transmembrane (TM) subunits, with a furin cleavage site (RXXR) between them [13,14]. In addition, there are two disulfide-bond isomerase motifs, Cys-X-X-Cys (CXXC) in SU and a CX6CC motif in TM, a putative receptor binding domain (RBD) in SU, an immunosuppressive domain (ISD) and two heptad repeats (HR1 and 2) in TM, all of which are well conserved. Importantly, *ERVPb1* is reported to be widely conserved in Catarrhini species, suggesting that it confers some evolutionary advantage. It is also reported that forced expression of the putative *ERVPb1* gene induces cell fusion in several types of cultured cells like *syncytin-1* and -*2*, even though it lacks a fusion peptide in TM [13,14]. *Syncytin-1* and -*2* exhibit placenta-specific expression, but *ERVPb1* mRNA expression is ubiquitous if also quite low in many tissues and organs, approximately 1/100 the level of *syncytin-1*, even in the placenta [13]. Therefore, it remains uncertain whether the ERVPb1 protein is actually translated during human development and, if so, in which lineage of cells it is functional.

Recently, various in vitro differentiation methods using mouse and human induced pluripotent stem cells (iPSCs) and/or embryonic stem cells have been developed for the generation of several different tissue and organ types, including the liver, brain, kidney, and heart as well as yolk sac-derived erythroid and myeloid cells as well as macrophages [15,16,17,18]. Such human organoid generation systems provide excellent models for mimicking human development and/or tissue differentiation, and are thus essential tools for elucidating the biological functions of primate- or human-specific *RVD* genes by means of gene targeting instead of having to employ knockout mice, because these genes are not present in mice. In this paper, we report a novel system for the highly sensitive detection of HERV-derived proteins/peptides in the process of tissue differentiation from human iPSCs. We obtained a human iPSC line with an insertion of a fluorescent Venus protein just after the N-terminal region of the ORF that was examined using CRISPR/Cas9 genome editing, and performed in vitro differentiation according to tissue organoid generation protocols. Using this system, we detected the ERVPb1–Venus fusion protein under a condition of macrophage differentiation from iPSCs, demonstrating that the ERVPb1 protein is actually translated in a hematopoietic cell lineage, including translation of a subset of macrophages.

## 2. Results

### 2.1. ERVPb1 Is Unique to Simiiformes and Exhibits a Quite Low Level of Expression in Limited Tissues and Organs in Humans

Previous reports indicated that *ERVPb1* is unique to Catarrhini [13,14], but its presence in the owl monkey was suggested by PCR using locus-specific intra-envelop primers [13]. In this study, we found that the orthologous sequence is also conserved in New World monkeys (Platyrrhini), such as marmosets, capuchins, and owl monkeys, but is absent from the Philippine tarsier, a Tarsiiforme species, and the greater bamboo lemur, a Strepsirrhine species, demonstrating that *ERVPb1* is conserved in both Catarrhini and Platyrrhini, so it is unique to Simiiformes (Figure 1a,b). *ERVPb1* is located between exons 4 and 5 (intron 4) of the *Ras* and *Rab* interactor 3 (*RIN3*) gene except in two cases: in gibbons between exons 5 and 6 due to a split of exon 4 (exons 4 and 5) and in African green monkeys (AGMs) between exons 3 and 4 due to a deletion of exon 1. This location is also conserved in the Platyrrhini species and its ORF, comprising 666 aa (660 aa due to a 6 aa deletion in marmosets), has been conserved in all of the ENV-like motifs and domains in the Catarrhine *ERVPb1* orthologues (Figure 1b). Thus, it is probable that the domestication of *ERVPb1* from the original HERVIP10B3-like retrovirus would have occurred before the Platyrrhini-Catarrhini split 44.2 MYA [19].

It is also reported that *ERVPb1* mRNA expression is ubiquitous and quite low in the tissues and organs in which it is found [13,14], approximately 1/100 the level of *syncytin-1*, even in the placenta [13]. We also examined mRNA expression by RT-PCR using a commercially available human multiple tissue RNA panel and confirmed that the mRNA level was quite low in human tissues and organs: 35 cycles of the RT-PCR reaction were required for detection in the thymus, lung, placenta, fetal liver, and bone marrow, with no expression detected in other tissues and organs (Figure 1c). These results were supported by the short read RNA-seq data of 26 human tissue and organ samples registered in the SRA database (SRA, https://www.ncbi.nlm.nih.gov/sra (accessed on 21 January 2012–22 March 2019)). The values of the reads per million (RPM) mapped reads were very low (Appendix A). There were a few brain samples (including the cerebrum, cerebellum, and hippocampus) and leukocyte cell samples (including B and T cells as well as macrophages) that exhibited > 0.1, but the placenta, heart, and most of the samples from the above tissues and cell types were nearly 0 (<0.1); the highest and second highest values were 0.76 and 0.33 from one hippocampus sample and one macrophage, respectively.

### 2.2. Generation of Human iPSCs with an ERVPb1–Venus Fusion Gene in Its Endogenous Locus

The above data indicate that the expression level of the putative ERVPb1 protein is quite low, even in the tissues and organs in which its mRNA expression was confirmed. Therefore, to determine the role of this protein in human development, we developed a novel detection system by combining a highly sensitive method for detecting target proteins by means of a protein fused with a fluorescent Venus protein (Irie et al. submitted) together with in vitro organoid formation using human iPSCs. We introduced the *Venus* gene into the endogenous *ERVPb1* in human iPS cells (HPS0063 201B7) using the CRISPR/Cas9 genome editing process so that the Venus protein was translated just 48 amino acids after the N-terminal of the putative ERVPb1 protein (Figure 2a). Therefore, it is expected that the ERVPb1–Venus fusion protein be translated by the same regulatory mechanism as the endogenous *ERVPb1* gene and thus detected in the tissues and organs with a high degree of sensitivity.

### 2.3. Detection of the ERVPb1–Venus Fusion Protein in a Hematopoietic Cell Lineage

The Venus knock-in human iPSCs were differentiated to see whether the putative ERVPb1 ORF is actually translated during the in vitro differentiation process of human iPS cells. First, we tested whether the Venus signal appeared in any of the cell types among the three germ layers differentiated from the embryonic bodies (EBs) of the Venus knock-in human iPSCs. We generated EBs in Cellartis DEF-CS 500 Culture System medium under feeder-free conditions for seven days, after which the EBs were differentiated into the three ectoderm, endoderm, and mesoderm germ cell lineages according to the Cellartis DEF-CS 500 Culture System in vitro differentiation protocol. After four days of differentiation, we detected the fluorescent signals emitted from the Venus fusion proteins. A strong Venus signal (with a maximal emission peak at 528 nm) was detected from one flattened cell as well as weak signals in a few other cell groups in a 3 cm diameter dish (Figure 2b,c). Although there were many other signals, they had peaks of 555, 565, 575, and 595 nm and were derived from the autofluorescence of the differentiated normal human iPS cells without any Venus insertion (Figure 2b). Therefore, it is critically necessary to distinguish the Venus-derived signal from the other autofluorescence signals. For this purpose, it is essential to use confocal fluorescence microscopy equipped with the function of Multi-channel Unmixing or Automatic Composition Extraction (ACE) (Figure 2b). From the cell morphology of the one flattened cell, it was determined to be a kind of macrophage, along with groups of cells that were hematopoietic cell precursors, such as hemangioblast cells.

We then differentiated the Venus knock-in human iPSCs according to the macrophage differentiation protocol using Cellartis DEF-CS 500 Culture System medium containing 80 ng/mL BMP4 without bFGF. Under this condition, many Venus-positive cells were induced after four days of culture (Figure 3a). We were also able to detect the Venus mRNA by RT-PCR, while it was hardly detected under the first differentiation condition (Figure 3b). As a control, human iPSCs without the Venus insertion were then differentiated using this protocol, and we were able to detect the *ERVPb1* mRNA by real-time PCR. The *ERVPb1* mRNA was detected in the period of days 3~5, with the peak on day 4 (Figure 3c,d), like the Venus signals from the Venus knock-in iPSCs (Figure 3b). Under this condition, the mesoderm marker *TBXT* was induced from day 1 of the differentiation process and peaked on day 2. The hemangioblast marker *CD235a* was detected on days 2 and 3 (Figure 3d), so the differentiation via the mesoderm into primitive hemangioblast cells occurred on these three days of differentiation. *ERVPb1* mRNA expression peaked on day 4 when the macrophage-like cells appeared in the differentiated Venus knock-in iPSC culture. Its expression was subsequently reduced, indicating that *ERVPb1* mRNA expression was temporary, that is, limited to the early stage of hematopoietic cell differentiation. These results clearly demonstrate that *ERVPb1* is a real, protein coding gene that manifests in hematopoietic cell lineages, including a subpopulation of macrophages, in human development. When we carried out the RT-PCR experiment using three primer sets, one designed within the ERVPb1 ORF and the other designed 28, 551 or 1487 bp upstream of the ORF, there was a band detected using the first primer set but not the last two primer sets, suggesting there is no full length mRNA (Appendix A). We then carried out the 5′-RACE experiment and found that the *ERVPb1* mRNA is transcribed from an unexpected promoter located approximately 2.5 kb upstream from the ORF, but not from the presumable 5′ LTR approximately 3.0 kb upstream, and then spliced to the 172 bp upstream of the first ATG codon of the ERVPb1 protein (Figure 3e).

Finally, we analyzed *ERVPb1* expression in primary human monocytes using the short-read RNA-seq data of 183 LPS-treated and 200 control samples registered in the European Genome–Phenome Archive (EGA: accession code EGA: EGAS00001001895) [20]. We mapped the single-end short-read RNA-seq data on the *ERVPb1* locus (Appendix A) and counted the data at normalized level. Interestingly, the *ERVPb1* expression significantly increased from 12.5099 to 16.2717104 reads with lipopolysaccharide (LPS) stimulation (Games–Howell test, significant, *p* = 8.71 × 10^−8^) (Figure 4). These data clearly demonstrate that *ERVPb1* is upregulated by LPS administration in primary human monocytes.

*ERVPb1* expression in primary human monocytes using short-read RNA-seq data of 183 LPS-treated and 200 control samples registered in the European Genome–Phenome Archive (EGA: accession code EGA: EGAS00001001895). Plots were constructed using the values of log2(count + 1). The *p*-value was calculated by two-sided Games–Howell test, which is parametric, assuming non-equal variance, and based on Welch’s statistics.

## 3. Discussion

It was demonstrated in an in vitro differentiation system using human iPSCs that *ERVPb1* is a bona fide protein-coding gene that is expressed in human hematopoietic cell lineages, including macrophages, and also that its mRNA is induced by LPS stimulation in primary human monocytes. Importantly, there is no full length mRNA, and only a spliced form of the *ERVPb1* mRNA is expressed in differentiated human iPSCs. Interestingly, it is not a typical spliced transcript from 5′ LTR as reported in the cases of the usual *ENV* and *ERVWE/syncytin-1* transcripts in seminomas [21,22], suggesting that it has a unique promoter that resides between the 5′ LTR and degenerated *GAG* sequence (Figure 3e). Thus, further analysis of the features of this unique promoter will be required in the future. It should be noted that the hemangioblast and hematopoietic cells induced in the in vitro human iPSC differentiation experiments have yolk sac-derived properties (Figure 2 and Figure 3). Thus, it is highly probable that *ERVPb1* is expressed in a subpopulation of both yolk-sac- and bone-marrow-derived hematopoietic cells, including macrophages. This explains why *ERVPb1* exhibits such a wide expression profile range in human tissues and organs. Yolk sac-derived macrophages become tissue-resident macrophages in certain tissues and organs, such as the lung, liver, kidney, epidermis, and brain (where they are known as microglia), and it is also possible that ERVPb1 is expressed in other lymphoid cell populations, such as B and T cells, based on the RNA-seq analysis data shown in Appendix A. Under normal conditions, *ERVPb1* expression is maintained at a low level, however, and importantly, its expression is induced by LPS stimulation in primary monocytes, and possibly by the administration of various interferons in other lymphoid cell lineages. Nevertheless, it remains elusive whether the cell fusion activity induced by its forced expression in a variety of cell lines is the actual function of the ERVPb1 in the course of normal human development [23]. The ERVPb1–Venus fusion protein lacks all of the other ENV-like motifs, but retains the N-terminal signal peptide motif, so the Venus signals were clearly detected in the Golgi body, a multi-layered structure in the proximity of the nucleus, in macrophages (Figure 3a). We expect that another type of fusion protein with a Venus protein in the C-terminus of ERVPb1 would prove useful for determining the precise cellular location of ERVPb1 in the macrophage cells, and would also be helpful in establishing the actual function of the ERVPb1 protein.

The presence of *ERVPb1* in Platyrrhini species as well as its absence from the Tarsiiforme as well as Strepsirrhini species demonstrates that its domestication from the original HERVIP10B3-like retrovirus must have occurred after the Tarsiiforme–Simiiforme split 71.1 MYA and before the Platyrrhini–Catarrhini split 44.2 MYA [19]. Its intact ORF has been conserved in the Simiiforme species, suggesting that it confers some evolutionary advantage. Human *syncytin-2* is also conserved in Simiiformes, but is derived from a different HERV (HERV-FRD) [9]. Thus, while the number of Simiiforme-specific *RVDs* is currently two, it is possible that more Simiiforme-specific *RVDs* may be found in the human genome if the HERV-derived sequences are searched with an ORF of less than 500 aa. Similarly, there may be more Catarrhini-specific genes in the human genome in addition to *syncytin-1* and *suppressyn*.

A series of KO mouse studies of mammalian-specific *RVD* genes, including *Peg10*, *Rtl1/Peg11*, and *Ldoc1/Sirh7/Rtl7*, as well as other *Rtl/Sirh* genes, has demonstrated that they exert important functions in the placenta and/or brain in mice and presumably humans [24,25,26,27,28,29,30]. Thus, the concept of mammalian-specific genes acquired from retroviruses is evidently well-supported, providing a useful model for approaching a variety of mammalian-specific features related to viviparous reproduction and the highly developed central nervous system in terms of genomic function. These results strongly indicate that these *RTL/SIRH* genes derived from retroviruses may be major players in the macroevolution of mammals [12,31,32]. It should be noted that most of them encode only a portion of the differently sized GAG-derived proteins, for example only 112 and 113 aa for RTL8C, A, B/SIRH4, 5, and 6; 151 aa for RTL7/SIRH7; 243 aa for RTL6/SIRH3; 304 aa for RTL4/SIRH11; and 599 aa for RTL5/SIRH8. Thus, not only ENV but also GAG and POL-derived ORFs may be functional in human development. There still remain more than 10,000 HERVs that have ORFs comprising >100 amino acids (aa) for GAG, POL, and ENV-derived proteins/peptides [3,4,33]. We have now reached a stage at which it is important to ask how many primate- and human-specific *RVD* genes exist in the human genome. It is critically important for understanding the evolutionary emergence of human beings to elucidate the biological functions of these genes during the course of human development and growth. Thus, speeding up the efforts to carry out a comprehensive screening of the primate- and human-specific *RVD* genes is of considerable importance.

## 4. Materials and Methods

### 4.1. Comparison of ERVPb1 ORF in Simiiformes

All of the HERV and HERV-derived ORFs as well as viral sequences were obtained from the Ensembl (hg38, version 103.38), Repbase (version 23.02), and gEVE databases (version 1.1), respectively [4,34]. Using the human ERVPb1 ORF sequence as a query, we conducted tblastn searches against the genome sequences of eight *Catarrhini* primates (bonobo, chimpanzee, gorilla, orangutan, gibbon, African green monkey, rhesus macaque, and baboon), three Platyrrhini primates (Ma’s night monkey, Capuchin and Marmoset), a Tarsiiforme species (Philippine tarsier), and a Strepsirrhini species (greater bamboo lemur). The amino acid identity in each of the ERVPb1 ORF sequences was calculated using the Clustal Omega program (Clustal Omega, https://www.ebi.ac.uk/Tools/msa/clustalo/ (accessed on 6 January 2021–10 April 2021)) in the default mode.

### 4.2. Analysis of Short Read RNA-Seq Data

The data in Appendix A are summarized for each SRA ID of the RNA-seq data obtained from the SRA database (SRA, http://www.ncbi.nlm.nih.gov/sra/ (accessed on 21 January 2012–22 March 2019)). They were aligned to HERV-derived ORFs in the human genome (hg38) obtained from the gEVE database (gEVE, http://geve.med.u-tokai.ac.jp (accessed on 18 February 2019)) using bowtie2 version 2.2.6 and the default parameter settings [35]. The expression level of each ORF was calculated by the total mapped reads as the reads per million (RPM) of the mapped reads. The RNA-Seq data of human primary monocytes of the 200 healthy individuals shown in Figure 4 were obtained from the European Genome–Phenome Archive (EGA), accession code EGA: EGAS00001001895), mapped onto the hg38 using STAR (version 2.73a) with the option “--outFilterMultimapNmax 1” and assembled using Scallop (version 0.10.5) [19,36,37]. The uniquely mapped read counts were normalized using size factors calculated in the DESeq2 package version 1.26.0 [38] and R version 3.6.1 [39].

### 4.3. Generation of Human iPSCs with an ERVPb1-NV Construct

Human ERVPb1-NV (Venus knock-in) iPSCs were generated with the CRISPR editing tool. A gRNA was designed to target the ERVPb1 N-terminal using the Benchling online tool (Benchling, https://www.benchling.com (accessed on 11 November 2019), then cloned into the pGuide-it-ZsGreen1 Vector (Guide-it™ CRISPR/Cas9 System, Takara Bio, Shiga, Japan). The gRNA oligonucleotide sequence was 5’-AGGTGGCTAGCAAGAGCACA. The ERVPb1–Venus NV donor vector was assembled with a 5′ homologous arm (1631 bp), Venus-Puro, and 3′ homologous arm (1651 bp) by means of a cloning enzyme (In-Fusion^®^ HD Cloning Kit, Takara Bio, Shiga, Japan). The plasmid containing the gRNA and the donor vector was introduced into human iPSCs by electroporation. Human iPSCs in which homologous recombination had occurred were selected by puromycin. ERVPb1-NV human iPSCs were confirmed by the DNA sequence from 5’HA to 3’HA.

### 4.4. Culture and Differentiation of Human iPSCs

Human iPSCs (HPS0063 201B7) were cultured in Cellartis DEF-CS 500 Culture System medium (Takara Bio, Shiga, Japan) under feeder-free conditions. Human iPSCs were induced into the embryoid body and differentiated into the three germ layers based on the in vitro differentiation protocol (Takara Bio). For macrophage differentiation, human iPSCs was induced in Cellartis DEF-CS 500 Culture System medium containing 80 ng/mL BMP4 (Recinbinant human BMP−4, R&D Systems, Minneapolis, MN, USA) without bFGF [17].

### 4.5. Gene Expression Analysis by Real-Time PCR

The total RNA of human tissues (Human Total RNA Master Panel II, Takara Bio) was used for RT-PCR. The RT-PCR primer used were *ERVPb1*-forward: 5′ GCTAGTAGGCCCTAATTGTTCAG, *ERVPb1*-reverse: 5′ AAGAAACCACAACGCCGTAA, *Venus*-forward: 5′ GTGAGCAAGGGCGAGGAGCTGTT, *Venus*-reverse: 5′ CTACTTGTACAGCTCGTCCA, *TBXT*-forward: 5′ GCACACCAGAATTTGTAAACTCC, *TBXT*-reverse: 5′ TGTCCTTAACAGCTCAACTC, *CD235a*-forward: 5′ TGTCCTTAACAGCTCAACTC, and *CD235a*-reverse: 5′ AAATATTCACCTGCATGTCC. The total RNA derived from human iPSCs was extracted using an RNeasy Plus Kit (QIAGEN, Hilden, Germany) and used to carry out first-strand cDNA synthesis using Superscript III (SuperScript™ III First-Strand Synthesis System, Thermo Fisher Scientific, Waltham, MA, USA). Real-time PCR was performed using PowerTrack SYBR Green Master Mix (Thermo Fisher Scientific) in a StepOne machine (StepOne Real-Time PCR System laptop, Thermo Fisher Scientific), technical triplicates were analyzed. The *ERVPb1*-forward and -reverse primers were used for real-time PCR. The PCR conditions were one cycle of 95 °C for 2 min, 40 cycles of 95 °C for 5 s, and 60 °C for 30 s. The relative expression level was determined by the standard curve method and normalized to the expression of *β-Actin*.

### 4.6. Rapid Amplification of cDNA Ends (RACE)

For the 5′-RACE experiment, the 5′-Full RACE Core Set (Takara Bio, Shiga, Japan) was used to extend the *ERVPb1* mRNA extracted in *4.5*. The first strand cDNA synthesis was carried out with 1 μg of total RNA using *ERVPb1* 5’-RACE GSP: 5’ TTATACATGTTCTGC (5′-phospholylated). The 5′-RACE PCR was performed with PrimeSTAR GXL DNA Polymerase (Takara Bio, Shiga, Japan) using the following primers: *ERVPb1*-race-F1: 5’ CAGGATGGTTAGGGCAATGG and *ERVPb1*-race-R1: 5’ GGTAATGGTCTGGGGTGAGT. The 5′-RACE PCR products (first 5′-RACE products) were confirmed by 1.5% agarose gel electrophoresis. Nested PCR was performed with PrimeSTAR GXL DNA Polymerase using the following primers: *ERVPb1*-race-F2: 5’ CGTGAACGAACACGAAATCCGAA and *ERVPb1*-race-R2: 5’ AGGTGGCTAGCAAGAGCACAA. The nested PCR products were purified by Agencourt AMPure XP (Beckman Coulter, Brea, CA, USA) and were determined by Sanger sequencing.

## 5. Conclusions

We have demonstrated that *ERVPb1* is a novel Simiiforme-specific gene and actually translated in hematopoietic cell lineages, including a subset of macrophages, in the differentiated human iPSCs. In addition, its expression is up-regulated by LPS administration in human primary monocytes.

## Figures and Tables

**Figure 1 ijms-22-04504-f001:**
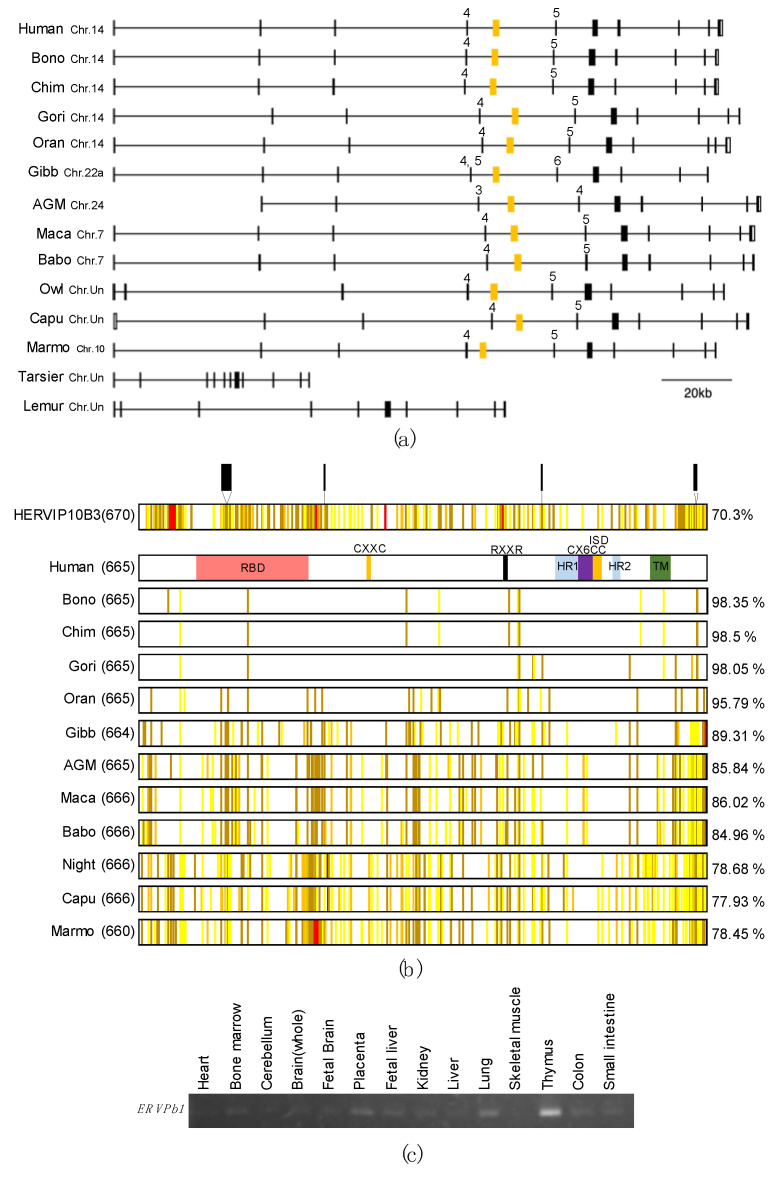
*ERVPb1* is conserved in Simiiformes and its mRNA expression is observed in human tissues. (**a**) Location of the conserved *ERVPb1* between *RIN3* exons 4 and 5 in Simiiformes including three Platyrrhini species. The orange and black colored boxes represent the *ERVPb1* locus and *RIN3* exons, respectively. Bono: Bonobo, Chim: Chimpanzee, Gori: Gorilla, Oran: Orangutan, Gibb: Gibbon, AGM: African green monkey, Maca: Macaca, Babo: Baboon, Owl: Ma’s night monkey, Capu: Capuchin, Marmo: Marmoset, Tarsier: Philippine tarsier, Lemur: Greater bamboo lemur. (**b**) The putative ERVPb1 ORFs found in Simiiformes and HERVIP10B3 ENV (upper line). The numbers in the parentheses and those on the right side indicate the number of amino acids and% identity to the human ORF, respectively. The black and red bars indicate insertion and deletion, respectively. The colored bars indicate conservation between groups of strongly similar (roughly equivalent to scoring > 0.5 in the Gonnet PAM 250 matrix: yellow) or weakly similar (roughly equivalent to scoring < 0.5 and > 0 in the Gonnet PAM 250 matrix: orange) properties, or that they are not similar (brown) to the human ORF. The colored boxes in the human ORF indicate ERVPb1 motifs and domains, such as CXXC and CX6CC: disulfide-bond isomerase motifs, HR1 and HR2: heptad repeats, ISD: immunosuppressive domain, RBD: receptor binding domain, RXXR: furin cleavage site, and TM: transmembrane domain. (**c**) *ERVPb1* transcripts detected by RT-PCR (270 bp) in human tissues. Thirty-five cycles of the PCR reaction were performed.

**Figure 2 ijms-22-04504-f002:**
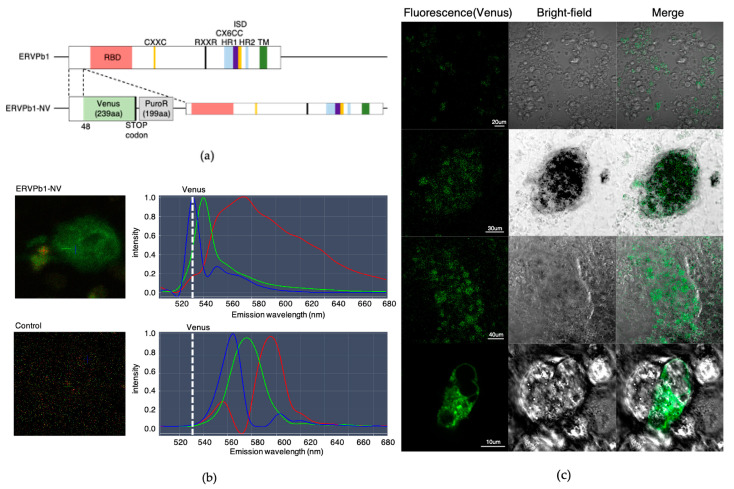
ERVPb1–Venus fusion protein detected under a condition of iPSC differentiation in the three germinal layers. (**a**) Structure of the ERVPb1-NV knock-in construct. The Venus gene was introduced into endogenous ERVPb1 so that the Venus protein was fused with ERVPb1 just after the 48 amino acids of its N-terminus (ERVPb1–Venus fusion protein). Most of the ENV features of this putative protein were removed in this construct except for the N-terminal signal peptide sequence. (**b**) Peak separation between the Venus-fusion protein (maximum peak emission fluorescence wavelength: 528 nm) and autofluorescence in human iPS cells with or without Venus insertion. The white dotted line indicates the wavelength of Venus. The ERVPb1–Venus fusion protein was detected only in the ERVPb1-NV iPSCs, not in the control iPSCs. (**c**) Fluorescence (Venus), Bright-field, and the merged images of ERVPb1-NV iPSCs differentiated into the three germ layers.

**Figure 3 ijms-22-04504-f003:**
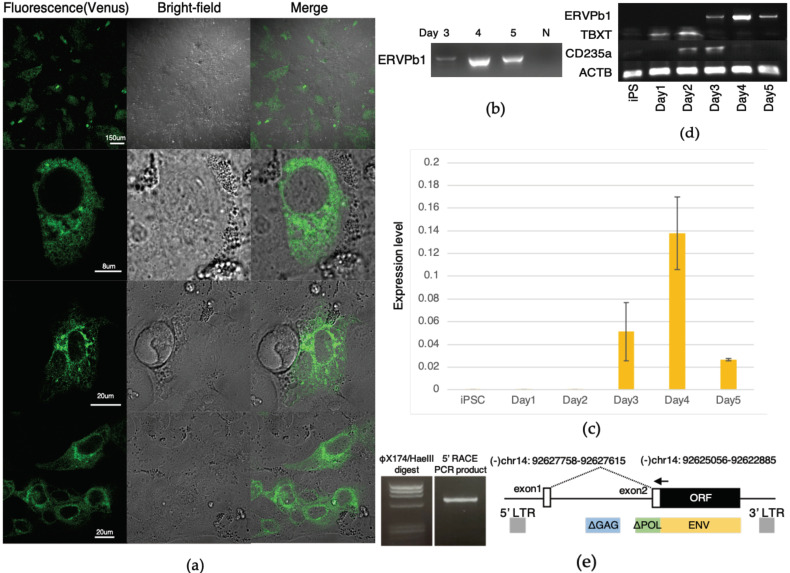
ERVPb1–Venus fusion protein detected under an iPSC differentiation condition induced in macrophages cells. (**a**) ERVP1-NV iPSCs were differentiated under a condition for macrophage differentiation containing BMP4 without bFGF in day 4 cultures. (**b**) Analysis of Venus expression by RT-PCR. (**c**) Detection of *ERVPb1* by real-time PCR in normal human iPSCs in the differentiation medium containing BMP4 on days 1–5. The values were normalized to *β-actin* expression and are shown in a bar graph. Each PCR was run in triplicate. (**d**) Detection of *TBXT* (a mesoderm marker) and *CD235a* (a hemangioblast marker) in addition to *ERVPb1* in normal human iPSC differentiation, the same as (**c**). (**e**) A spliced form of *ERVPb1* from an unusual promoter was detected in the differentiated human iPSCs using 5′-RACE. The arrow indicates PCR primer using 5′-RACE. The PCR product (535 bp) contained the known 219 bp sequence in the ERVPb1 ORF and upstream 316 bp sequences, which included the 144 bp exon 1 sequence approximately 2.5 kb upstream and a 172 bp sequence upstream of the first ATG in exon 2.

**Figure 4 ijms-22-04504-f004:**
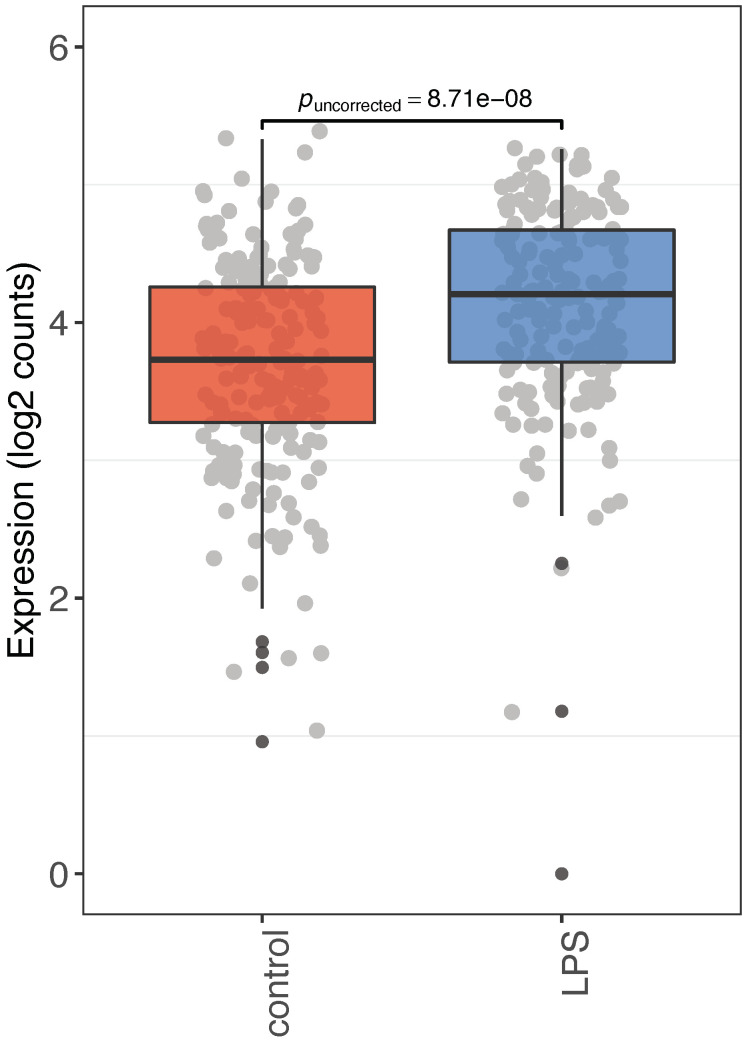
Increment of *ERVPb1* mRNA expression in primary monocytes by LPS administration. Comparison of RNA-Seq data from healthy primary and LPS-treated monocytes is presented.

## Data Availability

Data can be available upon request from the corresponding author.

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
