# Peer review of "HERV-Derived Ervpb1 Is Conserved in Simiiformes, Exhibiting Expression in Hematopoietic Cell Lineages Including Macrophages"

_ijms, 2021, doi:10.3390/ijms22094504_

Round 1

Reviewer 1 Report

Matsuzawa and coworkers present an elegant study of ERVPb1 gene expression during differentiation of induced pluripotent stem cells. Their data shed light on endogenous retrovirus derived Env-like gene. The experiments are sound, clearly presented. Simiiformes genome analysis would be worth to strengthen.

Major comment

The purpose of the analysis of simian genome is not described (line 99). It is unclear why the authors include a single platirrhini species. The novel findings would be better supported by analysis of several platirrhini species. Divergence between the findings and those published should be discussed.

Minor comments

Line 26 : “involved” should be replaced by “ expressed”

Line 27: iPSCs is not defined

Mentionning that Catarrhini species are old world monkey is useful for readers and should be mention in line 56 rather than in line 71.

Line 81: ES cells not defined

Author Response

Thank you very much for the reviewers’ important and critical comments and helpful suggestions. We addressed all the reviewers’ requests and we are sure that the revised manuscript is improved a lot.

Reviewer 1

Matsuzawa and coworkers present an elegant study of ERVPb1 gene expression during differentiation of induced pluripotent stem cells. Their data shed light on endogenous retrovirus derived Env-like gene. The experiments are sound, clearly presented. Simiiformes genome analysis would be worth to strengthen.

Major comment

The purpose of the analysis of simian genome is not described (line 99). It is unclear why the authors include a single platirrhini species. The novel findings would be better supported by analysis of several platirrhini species. Divergence between the findings and those published should be discussed.

Thank you for your point. We added the results of two other Platyrrhini species, capuchins and owl monkeys, then the total is now three. We also confirmed that there is no ERVPb1 in tarsiers (Haplorrhini) and lemurs (Strepsirrhini) in the RIN3 gene, thus, supporting our conclusion. We included this information in the abstract, results and discussion. We also added a citation of Aaggard et al. 2005 that they suggested the presence ofERVPb1 in owl monkey.

Minor comments

Line 26 : “involved” should be replaced by “ expressed”

We amended according to the reviewer’s request.

Line 27: iPSCs is not defined

We add full description of iPSCs.

Mentionning that Catarrhini species are old world monkey is useful for readers and should be mention in line 56 rather than in line 71.

We changed the position of the explanation.

Line 81: ES cells not defined

Here, we used “embryonic stem cells” and deleted “ES cells”, because we didn’t use this word afterward.

Reviewer 2 Report

In their manuscript “HERV-derived ERVPb1 is conserved in Simiiformes...“ Matsuzawa et al. identified ERV sequence orthologous to the human ERVPb1 in the genome of marmoset. This finding changes the chronology of ERVPb endogenization and sets the event by more than 10 MYA back before the split of Old World and New World monkeys. Second, authors fused the ERVPb1 env with Venus fluorescent reporter and detected specific signals of ERVPb1 expression in hematopoietic lineage, particularly in macrophages, differentiated from iPCS. This finding was corroborated by short-read RNA-seq analysis of primary human monocytes. Authors conclude that ERVPb1 plays a role in macrophage function/differentiation in addition to its previously suggested role in placenta development. Both findings are convincing and significant and I have just few concerns to be solved before further consideration of the manuscript.

There are multiple Platyrrhini species well covered with NGS genomic data and authors should seek for ERVPb1 sequences not only in marmosets but also in tamarins, capuchins, spider monkeys, and owl monkeys. The latest might be most interesting, because amplification of inner env sequence was mentioned already in Aagaard et al., 2005. I also suggest analysis of prosimian and tree shrew genomes, which might pose the ERVPb1 endogenization even deeper in primate evolution. Thanks to the conserved integration within the RIN3 gene, this analysis seems to be quite simple.

Second, do RNAseq data from primary human monocytes offer analysis of correct ERVPb1 splicing? This is important, because only spliced env mRNAs are translatable and the presence of longer transcripts might be irrelevant to env function (as shown in the case of aberrant syncytin-1 expression in cancer). Authors should clearly document the presence of spliced form of ERVPb1 mRNA. Alternatively, this question might be answered by splice-specific RT-PCR done with primary monocytes/macrophages.

Minor points:

In Fig. 1b, it is not clear for readers what is the difference between HERVIP10B3 and Human and what represent the colored bars in HERVIP10B3. Also, env domain structure could help in understanding the interspecific conservation. In Fig. 1c, description of the indicative RT-PCR product is strange.

In Fig. 2a, the Venus-ERVPb1 fusion deserves more detailed scheme because it is substantial piece of work and reporter knock-in technology is not trivial.

Author Response

Thank you very much for the reviewers’ important and critical comments and helpful suggestions. We addressed all the reviewers’ requests and we are sure that the revised manuscript is improved a lot.

Reviewer 2

In their manuscript “HERV-derived ERVPb1 is conserved in Simiiformes...“ Matsuzawa et al. identified ERV sequence orthologous to the human ERVPb1 in the genome of marmoset. This finding changes the chronology of ERVPb endogenization and sets the event by more than 10 MYA back before the split of Old World and New World monkeys. Second, authors fused the ERVPb1 env with Venus fluorescent reporter and detected specific signals of ERVPb1 expression in hematopoietic lineage, particularly in macrophages, differentiated from iPCS. This finding was corroborated by short-read RNA-seq analysis of primary human monocytes. Authors conclude that ERVPb1 plays a role in macrophage function/differentiation in addition to its previously suggested role in placenta development. Both findings are convincing and significant and I have just few concerns to be solved before further consideration of the manuscript.

There are multiple Platyrrhini species well covered with NGS genomic data and authors should seek for ERVPb1 sequences not only in marmosets but also in tamarins, capuchins, spider monkeys, and owl monkeys. The latest might be most interesting, because amplification of inner env sequence was mentioned already in Aagaard et al., 2005. I also suggest analysis of prosimian and tree shrew genomes, which might pose the ERVPb1 endogenization even deeper in primate evolution. Thanks to the conserved integration within the RIN3 gene, this analysis seems to be quite simple.

Thank you very much for your comments. We searched the data on other New World monkeys. We confirmed the presence of RIN3 and ERVPb1 genes in capuchins and owl monkeys in addition to marmosets. However, there is no information of RIN3 in tamarins and spider monkeys presumably because of shortage of sequence data. We also found the RIN3 gene in tarsiers (Haplorrhini), lemurs (Strepsirrhini) and treeshrews but no ERVPb1 in the RIN3 gene in these species. Then, we added information of the former two species into Fig. 1 to support our conclusion that ERVPb1 is Simiiforme-specific.

Second, do RNAseq data from primary human monocytes offer analysis of correct ERVPb1 splicing? This is important, because only spliced env mRNAs are translatable and the presence of longer transcripts might be irrelevant to env function (as shown in the case of aberrant syncytin-1 expression in cancer). Authors should clearly document the presence of spliced form of ERVPb1 mRNA. Alternatively, this question might be answered by splice-specific RT-PCR done with primary monocytes/macrophages.

Thank you for your critical comment. As there was no splicing information on the RNAseq data, we carried out RT-PCR and 5’-RACE experiments using differentiated human iPSCs to answer this question. Importantly, there was only a spliced form of ERVPb1 mRNA from an unexpected promoter between 5’ LTR and degenerated GAG, but no full length mRNA was detected. We added these data in Fig. 3e and Supplementary Fig. 1 and clearly documented in the text.

Minor points:

In Fig. 1b, it is not clear for readers what is the difference between HERVIP10B3 and Human and what represent the colored bars in HERVIP10B3. Also, env domain structure could help in understanding the interspecific conservation.

We added the explanation of the colored bars in HERVIP10B3, representing how similar or different properties the mutated amino acids have. We also added explanation of the domains and motifs of ERV in the human ERVPb1. So, we believe that it becomes easy to understand the interspecific conservation and diversification of ERVPb1 compared to the human ERVPb1.

In Fig. 1c, description of the indicative RT-PCR product is strange.

We changed the description according to the reviewer’s comment.

In Fig. 2a, the Venus-ERVPb1 fusion deserves more detailed scheme because it is substantial piece of work and reporter knock-in technology is not trivial.

We exchanged the figure for easy understanding the construction of Venus knock-in iPSCs.